# The Role of Systemic Microvascular Dysfunction in Heart Failure with Preserved Ejection Fraction

**DOI:** 10.3390/biom12020278

**Published:** 2022-02-09

**Authors:** Jerremy Weerts, Sanne G. J. Mourmans, Arantxa Barandiarán Aizpurua, Blanche L. M. Schroen, Christian Knackstedt, Etto Eringa, Alfons J. H. M. Houben, Vanessa P. M. van Empel

**Affiliations:** 1Department of Cardiology, CARIM School for Cardiovascular Diseases, Maastricht University Medical Centre (MUMC+), 6229 HX Maastricht, The Netherlands; sanne.mourmans@mumc.nl (S.G.J.M.); a.barandiaran@mumc.nl (A.B.A.); b.schroen@maastrichtuniversity.nl (B.L.M.S.); c.knackstedt@mumc.nl (C.K.); vanessa.van.empel@mumc.nl (V.P.M.v.E.); 2Department of Physiology, CARIM School for Cardiovascular Diseases, Maastricht University, 6211 LK Maastricht, The Netherlands; e.eringa@maastrichtuniversity.nl; 3Department of Physiology, Amsterdam Cardiovascular Sciences, Amsterdam University Medical Center, 1105 AZ Amsterdam, The Netherlands; 4Department of Internal Medicine, CARIM School for Cardiovascular Diseases, Maastricht University Medical Centre (MUMC+), 6229 HX Maastricht, The Netherlands; b.houben@maastrichtuniversity.nl

**Keywords:** heart failure with preserved ejection fraction, microcirculation, microvascular dysfunction, endothelial dysfunction

## Abstract

Heart failure with preserved ejection fraction (HFpEF) is a condition with increasing incidence, leading to a health care problem of epidemic proportions for which no curative treatments exist. Consequently, an urge exists to better understand the pathophysiology of HFpEF. Accumulating evidence suggests a key pathophysiological role for coronary microvascular dysfunction (MVD), with an underlying mechanism of low-grade pro-inflammatory state caused by systemic comorbidities. The systemic entity of comorbidities and inflammation in HFpEF imply that patients develop HFpEF due to systemic mechanisms causing coronary MVD, or systemic MVD. The absence or presence of peripheral MVD in HFpEF would reflect HFpEF being predominantly a cardiac or a systemic disease. Here, we will review the current state of the art of cardiac and systemic microvascular dysfunction in HFpEF (Graphical Abstract), resulting in future perspectives on new diagnostic modalities and therapeutic strategies.

## 1. Introduction

Heart failure with preserved ejection fraction (HFpEF) is a health care problem of epidemic proportions, currently accounting for roughly 3 million patients in the United States alone [1,2]. Over 50% of all patients with heart failure (HF) suffer from HFpEF, and its incidence is increasing by 1% each year [1]. This rising incidence parallels increasing rates of comorbidities and age. However, HFpEF is more than just a correlate of comorbidities [3]. Rather, HFpEF seems to be driven by them [4]. Moreover, growing evidence suggests HFpEF is a heterogeneous syndrome comprising of different phenotypes [5].

The paucity of effective treatments for HFpEF creates an urge to better understand the pathophysiology of this condition. Important comorbidities associated with HFpEF such as obesity, diabetes mellitus, renal dysfunction and hypertension have been linked with a systemic-low grade pro-inflammatory state [6]. This pro-inflammatory state is associated with increased oxidative stress and reduced nitric oxide (NO) bioavailability in endothelial cells (ECs), marking an EC phenotype shift towards an activated, pro-inflammatory state [7,8]. It has been postulated that, as a consequence, abnormal remodelling occurs (cardiomyocytes hypertrophy and stiffen, and local fibroblasts are activated to produce collagen), ultimately leading to diastolic dysfunction and HF [7].

Both coronary MVD and peripheral MVD, in this review defined as MVD in vascular beds other than the heart, have previously been reported in HFpEF [9,10,11,12,13,14,15,16]. These findings raise the question of whether either isolated coronary MVD or a more generalized, systemic MVD, consisting of both coronary and peripheral MVD, contribute to developing HFpEF. In spite of that, MVD is not limited to HFpEF. Comorbidities highly prevalent in HFpEF, such as hypertension, diabetes mellitus, and obesity, are also associated with MVD [17]. The reported associations between both HFpEF and its comorbidities with MVD complicate the discussion of whether MVD is a cause of HFpEF or a bystander.

In this narrative review, we will focus on current evidence and future perspectives of microvascular studies in HFpEF patients and review evidence for the involvement of systemic MVD in its pathogenesis. Improved understanding of the (patho)physiological relevance of peripheral MVD for HFpEF, paralleled by improved understanding of the links between systemic and coronary MVD, will aid the search for new diagnostic modalities as well as novel therapeutic targets [18].

## 2. Defining Microvascular Function and Dysfunction

Adequate interpretation of HFpEF studies that focus on MVD requires knowledge of the microcirculation and its functions from the vessel network and vessel type to a molecular level. Interpretation is complicated by the absence of a gold standard to define and diagnose microvascular dysfunction. Rather, different techniques evaluate different functional and structural aspects of the microcirculation and in different tissues, and a variety of these have been used in clinical research in HFpEF. The techniques have been reviewed elsewhere [19], and for clarity a summary can be found in Appendix A.

Microcirculatory beds are highly dynamic networks that constantly adapt to a variety of systemic and local signals from surrounding tissue and the vascular lumen, acting both acutely and chronically [20]. These signals include humoral, physical, neurogenic, cellular, and metabolic factors [20]. The primary function of the microcirculation is to meet demands for delivery of nutrients including oxygen to local tissue through flow regulation (mainly through regulation of vascular tone), structural adaptation (such as angiogenesis or rarefaction), permeability, haemostasis, immunity, and inflammation [21]. The way a microvessel functions differs per vessel type (arteriole, capillary, venule), organ, and position in the highly heterogeneous vascular tree [22]. These factors all contribute to different responses to local haemodynamics, rheology, and signalling factors/metabolites. Figure 1 displays a simplified example of the intricate interplay between different cell types in an arteriole and alterations that have been reported in HFpEF patients. The following sections will elaborate on the microvascular alterations found in prior research.

MVD is often used as a broadly defined term encompassing all aspects of abnormal functioning of the microcirculation. MVD is not a synonym for endothelial dysfunction since it is not limited to functional or structural alterations of ECs but can include any (cellular) component of the microcirculation such as smooth muscle cells (SMCs), matrix, or pericytes. Here, we define microvascular function as a continuum between normal function and dysfunction, instead of a binary phenomenon, and microvascular dysfunction as a state where the primary microvascular functions are suboptimal and affect the surrounding tissue.

## 3. Evidence of Microvascular Dysfunction in HFpEF

Since Paulus and Tschöpe postulated a central role for MVD in the aetiology of HFpEF, a substantial number of studies have investigated MVD in HFpEF [7]. At first, the focus was on coronary MVD, also referred to as coronary microvascular dysfunction (CMD), but with the understanding that HFpEF is a condition related to systemic comorbidities such as hypertension and diabetes mellitus, the involvement of systemic MVD was proposed [15,16]. In this section, we will review evidence of systemic MVD in HFpEF as defined by different microvascular functions. These include vasoreactivity, the response of vascular tone to an external stimulus, and capillary rarefaction, a reduction in the capillary density within tissues. Evidence is further grouped according to the location of the microvascular bed. Data are mainly available on different microvascular abnormalities throughout the body as correlates of HFpEF. An overview of the studies on HFpEF performed to date, to the best of our knowledge, is provided in Table 1 for peripheral MVD studies and in Table 2 for coronary MVD studies. 

In addition to the various different techniques and stimuli used to assess MVD in HFpEF, an important caveat concerning the evidence relating MVD to HFpEF (Table 1 and Table 2) is the use of various diagnostic criteria for HFpEF. For example, LVEF > 40% vs. >50% was used, ambulant vs. recently hospitalized patients, and use of different brain natriuretic peptide cut-off values. This diversity leads to different patient selection in terms of disease stage and clinical phenotype [5] and is summarized in Appendix A. Although the diversity resembles current clinical practice for HFpEF diagnosis and management [23], it complicates the integration of the clinical datasets.

### 3.1. Vasoreactivity in HFpEF

To study MVD in HFpEF, most publications have focused on using various external stimuli to influence vascular tone (vasodilation or vasoconstriction) of microvascular beds, known as vasoreactivity (Table 1 and Table 2). Vasoreactivity primarily reflects endothelial and/or SMC function and has specific responses to different stimuli, classified as endothelium-dependent and endothelium-independent. These categories, however, are not fully mutually exclusive, as most of the current techniques do not show cell type-specific vasodilation due to the dynamic nature of the microcirculation in vivo. For instance, adenosine is an endothelium-independent vasodilator [42], but its responses in vivo are also modulated by flow-dependent NO production of endothelial cells [43,44]. Moreover, supra-physiological adenosine concentrations, as used in coronary tests, are used to assess maximal hyperaemia; increased blood flow as a consequence of maximal vasodilatation [45]. Maximal hyperaemia is influenced by structural microvascular properties such as rarefaction as well as endothelium-independent vasodilation. In addition, stimuli that induce tachycardia also influence blood flow through active hyperaemia [46,47]. Findings of normal maximal hyperaemia do not exclude impairment of active hyperaemia under physiological conditions due to the involvement of more processes that could compensate for the processes occurring under physiological conditions [48]. Nevertheless, the classification of endothelium-dependent and independent vasoreactivity, as used in this review, can help to identify different underlying mechanisms for MVD or disturbed downstream pathways.

#### 3.1.1. Endothelium-Dependent Vasodilation

Impaired endothelium-dependent vasodilation and hyperaemia of peripheral microvascular beds have been studied scarcely in HFpEF. In a subgroup of coronary artery disease patients with HFpEF, HF with reduced ejection fraction (HFrEF), and no HF, one small study featured laser Doppler imaging of the forearm skin blood flow coupled with transcutaneous iontophoresis (delivery of a substance through the skin using a small electric current) using acetylcholine (endothelium-dependent) and sodium nitroprusside (endothelium-independent) [11]. Impaired endothelium-dependent and endothelium-independent hyperaemia were observed in both HFpEF and HFrEF patients compared to controls [11]. Another study reported impaired endothelium-dependent vasomotion measured as a rhythmic variation of blood flow using laser Doppler flowmetry in HFpEF patients compared to hypertensive controls [30]. However, the small sample size of the studies and the inadequate matching of comorbidities in the control groups limit the generalization of these results. Since all patients in the first mentioned study had a form of coronary artery disease [11], the findings of this study could mainly be driven by triggers involved in atherosclerosis [49] rather than HF(pEF) [50], although similar pathways have been identified [49,50].

In the heart, impaired coronary endothelium-dependent vasoreactivity has been demonstrated in HFpEF [9,32,33]. Yang et al. showed impaired coronary microvascular function in 72% of HFpEF patients who underwent invasive coronary microvascular assessment. This entailed 29% of the patients with isolated impaired endothelium-dependent coronary vasoreactivity measured by coronary blood flow (CBF, cut-off ≤ 0% increase) using intracoronary acetylcholine infusion; isolated impaired coronary maximal hyperaemia in 33% by coronary flow reserve (CFR, cut-off < 2.5) using adenosine infusion; and combined impaired coronary vasoreactivity in only 10% (both CFR < 2.5 and CBF ≤ 0%) [9]. Similar findings were confirmed in hospitalized HFpEF patients using invasive measurements [33]. Those results suggest that impaired coronary vasoreactivity is underreported when only endothelium-dependent vasodilatation is assessed [9,33]. In contrast, another recent study similarly used CFR and CBF and reported more endothelium-dependent MVD (86%) than impaired maximal hyperaemia (46%) in HFpEF patients, which were both more prevalent compared to controls without HFpEF (35% and 21%, respectively) [32]. Moreover, at rest, endothelium-dependent MVD, but not impaired maximal hyperaemia, was associated with higher cardiac filling pressures, a sign of cardiac congestion and a hallmark of HFpEF [1]. During exercise, both forms of MVD showed this association [32]. These results could, on the one hand, reflect that some HFpEF patients show different coronary microvascular alterations as compared to other HFpEF patients within the same population and compared to other populations, which could represent unequal disease progression or different underlying MVD mechanisms within the broad HFpEF spectrum. On the other hand, interpretation of differences between study results is limited by the low sample size and inadequate correction for confounders, absence of a control group, or a retrospective design that take in a selection bias. This bias encompasses that included patients have a higher likelihood of having coronary abnormalities than a general HFpEF population because of clinical suspicion of coronary abnormalities being the indication for the diagnostic test.

A promising vascular bed to provide knowledge on endothelium-dependent and independent MVD in HFpEF is the retina, mainly because the retina allows the evaluation of both structural and functional microvascular features by direct imaging of microvessels [51,52]. Studies have suggested a link between alterations in the retinal microcirculation and HFpEF; an association was shown between retinal microvascular changes, i.e., decreased arteriolar calibres (vessel widths) and increased venular calibres, and increased LV concentric remodelling [53,54], one of the characteristics for HFpEF [3]. Moreover, it was shown that retinopathy and widening of retinal venular calibres, but not narrowing of arteriole calibres, independently predicted HF incidence in large datasets [54,55]. Finally, an association was found between current and incident HFpEF and (self-)reported retinopathy as a complication of diabetes mellitus [56,57], as well as neuropathy and nephropathy [56]. This reported association was stronger than the association with HFrEF [56]. However, retinal alterations are associated with many other factors in similar datasets [58]. Thus, retinal microvascular alterations are presumably present in HFpEF, but which specific alterations are present in HFpEF remain to be identified.

Several other techniques have been used more frequently to assess endothelium-dependent and endothelium-independent flow responses in HFpEF, including flow-mediated dilation (FMD) of larger arteries [59]. As those studies have focused on the macrocirculation and its associated endothelial phenotype [22,60], this evidence falls outside of the scope of this review but has been reviewed elsewhere [61].

Furthermore, reactive hyperaemia by peripheral arterial tonometry of the finger is often referred to as microvascular endothelial dysfunction in studies but is mostly investigated by vasoactive stimuli that are not endothelium specific. Data from this technique are presented in the next section, while we provide more context on the technique here. This assessed reactive hyperaemia resembles post-ischemic hyperaemia after 5 min arterial occlusion involving many microvascular metabolites, including adenosine and NO, rather than only endothelium-dependent hyperaemia, which would occur after 1–3 min occlusion [62]. This response being not fully endothelium-specific is further supported by studies showing poor comparability of results of digital post-occlusive hyperaemia with other microvascular assessments such as laser Doppler flowmetry with iontophoresis of an endothelium-dependent stimulus [63,64]. Moreover, the reactive hyperaemia is minimally affected after smoking cessation [65,66], while smoking is a well-known trigger for oxidative stress and its effects on NO-dependent endothelial vasodilatation [67,68]).

#### 3.1.2. Endothelium-Independent Vasodilation

Peripheral endothelium-independent vasodilation has been studied in HFpEF predominantly in the digital microcirculation. Following the reported association of post-occlusive reactive hyperaemia in the finger using peripheral arterial tonometry with multiple cardiovascular risk factors in the Framingham Heart study [69], most HFpEF studies have used this technique. Several studies reported a lower hyperaemic response after 5 min occlusion of the brachial vasculature in HFpEF patients compared to matched control subjects [24,26,28,29], which was associated with worse outcome [25]. However, HFpEF patients showed a better digital hyperaemic response than HFrEF patients [27]. Impaired endothelium-independent vasodilation was also shown in the forearm skin in coronary artery disease patients with HFpEF compared with patients without HF [11]. Nonetheless, multiple comorbidities have been associated with MVD regardless of HF (Table 3) [19,70,71,72]. Therefore, it is important to take comorbidities into account when interpreting these results. Some studies were small, and the control individuals were often inadequately matched for comorbidities (Table 1 and Appendix A) [11,28,29]. Controls matched for hypertension showed worse digital hyperaemia compared to healthy controls and showed no difference compared to HFpEF in one study [29], but in two other larger studies, HFpEF subjects had worse digital hyperaemia compared to controls matched for at least hypertension, age, sex, and diabetes mellitus [24,26]. These data underline that MVD is influenced by multiple clinical factors, but particularly show that HFpEF seems to have exaggerated MVD compared to its comorbidities.

Clinical studies focusing on endothelium-independent vasodilation of the coronary microcirculation have assessed hyperaemia using different stimuli. Invasive studies have revealed impaired maximal hyperaemia in HFpEF patients compared to controls without HF, both free of epicardial stenosis. In particular, these studies reported impaired coronary flow reserve (CFR) and an increased index of microcirculatory resistance (IMR) after intracoronary adenosine infusion [13,14,32,34]. In parallel, non-invasive imaging studies assessing coronary vasoreactivity using positron emission tomography (PET), magnetic resonance imaging (MRI), or echocardiography revealed similar results, mainly using adenosine for maximum hyperaemia and dipyridamole or dobutamine as a stimulus for active hyperaemia [35,36,37,38,41] (Table 2). Moreover, impaired maximal hyperaemia was associated with more adverse events in HFpEF [39]. All but two of these coronary MVD studies were retrospective and, thus, also take in a selection bias with a higher likelihood of having coronary abnormalities than a general HFpEF population. In addition, the controls in these studies had fewer comorbidities than HFpEF patients, and limited confounder correction was performed. Nevertheless, the results all pointed in the same direction; coronary MVD is present in HFpEF.

To date, only one study assessed both coronary and peripheral (micro)vascular function in HFpEF. In their assessment, Shah et al. used novel echocardiographic tools (Doppler imaging to assess CFR of the left anterior descending artery after adenosine infusion) and observed a high prevalence of impaired maximal coronary hyperaemia in HFpEF (75% of 202 patients, no controls). This was significantly but weakly correlated with impaired peripheral vasoreactivity as measured by endoPAT [10]. This direct comparison between the peripheral skin microvasculature and coronary microvasculature is particularly limited by the use of different triggers to assess vasoreactivity (adenosine vs. ischemia), which would already result in different hyperaemic responses in the same vascular bed [62].

Taking all published results regarding vasoreactivity of different (micro)vascular beds together, HFpEF patients consistently show impaired (micro)vascular vasoreactivity throughout the body, suggestive of systemic MVD. However, not all patients show identical MVD phenotypes. Nonetheless, a causative conclusion on the gradual development of MVD and HFpEF requires more robust evidence.

### 3.2. Capillary Rarefaction in HFpEF

Capillary rarefaction in HFpEF has been shown in vascular beds of the upper legs and the heart with tissue biopsies [12,16]. By performing histology on biopsy samples of the upper leg, Kitzman et al. have shown that capillary rarefaction is present in skeletal muscle of HFpEF patients compared to age-matched controls and is associated with a reduced exercise tolerance [16]. Capillary rarefaction could be one of the factors that limit peak exercise due to impaired nutrient and oxygen supply to muscle cells [97]. Subsequently, abnormal mitochondrial oxidative capacity of skeletal muscle cells was found in upper leg biopsies of HFpEF patients compared to sedentary healthy controls [97]. Other studies reported abnormal skeletal muscle oxidative metabolism in HFpEF using maximal exercise testing and phosphate magnetic resonance spectroscopy (P-MRS) (upper leg) [98], near-infrared spectroscopy (upper leg) [31] or invasive hemodynamic monitoring [99], which seemed to be mainly due to impaired peripheral oxygen extraction. One study reported impaired skeletal muscle oxidative metabolism in HFpEF patients compared to controls with similar whole-body blood flow and indices of cardiac reserve (peak cardiac output and stroke work) [98]. Although these studies consistently report impaired peripheral oxygen extraction, conclusions are limited due to the considerable small sample sizes (<25 HFpEF patients) [16,31,97,98]. An impaired oxidative function could be caused by abnormalities within the muscle cells or oxygen supply via the microcirculation. Hence, these data suggest that both abnormal muscle cell metabolism and capillary rarefaction of the legs are present in HFpEF. Both could play a major role in the exercise capacity of HFpEF patients.

Coronary capillary rarefaction in HFpEF has been studied scarcely. Mohammed et al. assessed coronary capillary density by histology in autopsy samples and found that coronary rarefaction was more prevalent in HFpEF patients compared to age-matched controls. In addition, it was correlated with increased myocardial fibrosis [12]. This singular study is of importance, as it showed the structural coronary microvascular abnormalities (capillary rarefaction) in HFpEF. However, this study was performed post-mortem, and the included patients were likely to be in more advanced stages of HFpEF. Therefore, the question remains to what extent these structural abnormalities are present in HFpEF patients at an earlier stage, and whether these abnormalities are a cause of advanced HF or a consequence. For example, reduction of cardiac output can impair organ perfusion and myocardial function through neurohumoral activation [100,101]. Two imaging studies provide indirect information on cardiac capillary rarefaction, although findings are not consistent and controls had considerably fewer comorbidities, including less diabetes mellitus. One study used cardiac phosphorus magnetic resonance spectroscopy (P-MRS) to show impaired cardiomyocyte oxidative metabolism at rest in HFpEF patients compared to healthy controls [102]. This could occur due to myocardial mitochondrial dysfunction or due to capillary rarefaction. In contrast, the other study used PET with a dobutamine stress test to depict an increased rather than impaired myocardial oxygen extraction and blood flow at rest in HFpEF patients compared to controls matched for age and sex. However, HFpEF patients did show a blunted dobutamine-induced increase in myocardial oxygen extraction and blood flow compared to controls [35], which could occur due to impaired active hyperaemia as a consequence of rarefaction [103]. The inconsistent findings at rest could be explained by the widely accepted assumption that patients in an early stage of HFpEF only show impairments during exercise [1]. The first study had stricter inclusion criteria of impaired peak exercise in [102], possibly leading to the inclusion of patients with a worse clinical status and a more progressed HF stage compared to the second study [35]. Regardless, evidence of coronary capillary rarefaction in HFpEF remains scarce.

### 3.3. Microvascular Biomarkers in HFpEF

Serum endothelial biomarkers predominantly represent the microcirculation, considering that 98% of all ECs are part of the microcirculation [104]. Research on serum biomarkers and HFpEF is an evolving field, which has been reviewed several times before [105,106]. Here, we will briefly recap biomarkers that represent the microvascular function. 

Endothelial leukocyte adhesion molecules seem to be the most promising markers of general microvascular function and could be used to assess the sequelae of systemic MVD longitudinally. Intercellular adhesion molecule 1 (ICAM-1), vascular cell adhesion molecule 1 (VCAM-1), and E-selectin are responsible for leukocyte adhesion to ECs [107]. Their expression is upregulated when ECs are pro-inflammatory, including when MVD is present. Indeed, ICAM-1, VCAM-1, and E-selectin were upregulated in myocardial biopsies of HFpEF patients compared to controls [8,108]. Furthermore, elevated serum levels of E-selectin and ICAM-1 in younger adults were associated with future decreased systolic dysfunction (LV global longitudinal strain) while LVEF was preserved [109] and increased VCAM-1 levels were associated with incident HFpEF over a median follow-up of 14 years [110], suggesting that changes in microvascular function can be seen even before a clinical stage of HF occurs. 

Elevated plasma levels of von Willebrand Factor (vWF) were shown to be of incremental prognostic value on top of routine clinical characteristics and biomarkers in HFpEF [111], which were also associated with elevated adhesion molecules. vWF is associated with microvascular function, albeit this association is less convincing than adhesion molecules [112]. vWF is released by endothelial injury and plays a complex role in flowmotion due to its large structure. It mainly plays a role in haemostasis, in which vWF mediates platelet adhesion and interaction to the endothelium [112]. Hence, vWF might play a role in systemic MVD identification in HFpEF, but endothelial adhesion molecules seem to be more promising.

In general, there is much interest in microRNAs as biomarkers of microvascular or endothelial dysfunction, whether in blood or urine, encapsulated in exosomes or other types of vehicles [113,114]. MicroRNAs could have prognostic value as well because of their involvement in specific microvascular functions. For instance, microRNAs miR-150-5p, miR-21-3p and miR-30b-5p have been found in extracellular vesicles obtained from patients with diabetic nephropathy [115], a condition that impairs endothelial control of cardiomyocyte function [116]. These microRNAs were found to be involved in abnormal angiogenesis and therefore suggested to have potential as prognostic biomarkers of microvascular dysfunction in these patients. Studies on circulating microRNAs in HFpEF have predominantly focused on discriminating HFpEF from HFrEF [117], but have not (yet) yielded a candidate with sufficient discriminative value. Nevertheless, X-chromosome-linked microRNAs associated with microvascular dysfunction, such as miR-244 and -452 [118], may improve the understanding of sex-specific mechanisms in HFpEF [119], although the clinical significance of their circulating levels in relation to LV diastolic dysfunction and HFpEF is incompletely understood. Characterizing HFpEF-associated MVD in more detail with microRNAs as biomarkers could yield more potent and function-specific novel therapeutic targets. For this, studies are required to identify specific characteristics of MVD in HFpEF patients, which is in line with the plea of the current review.

## 4. Other Tissue Alterations and Pathways Related to MVD in HFpEF

### 4.1. Adipose Tissue Changes and MVD

Alterations in both cardiac and peripheral adipose tissue are present in HFpEF [74,120,121,122,123,124,125], including changes in the release of metabolites, cytokines, chemokines, and oxygen radicals [124,126]. Consequently, changes in adipose tissue signalling can affect local inflammation, atherosclerosis, and microvascular function [126,127]. In isolated vessels, it has been shown that microvascular function in muscle is controlled by perivascular adipose tissue (PVAT) [128,129] and is disrupted by inflammation of this tissue [130,131,132]. This PVAT control of microvascular and muscle function has also been shown in vivo, albeit not in relation to inflammation [133]. Studies on PVAT in HFpEF have not been reported, but enlarged adipose tissue depots have, and these can be associated with PVAT alterations, particularly where adipose tissue is in close proximity to microvessels such as the heart [134]. 

Excess epicardial adipose tissue (EAT) was found in HFpEF patients compared to HFrEF and healthy controls [135,136], which was associated with worse exercise capacity and worse survival [135]. Higher cardiac filling pressures were also found in obese HFpEF patients with excess EAT compared to those without excess EAT [125]. The strong influence of EAT on local cardiac structure and function is further supported by a recent cardiac MRI study in a mixed HFpEF/HF cohort with mildly reduced ejection fraction (HFmrEF) [137]. In addition to excess EAT, HFpEF patients showed several enlarged adipose tissue depots (visceral and intramuscular) compared to controls, and these enlarged depots were associated with impaired exercise tolerance [74,136] and higher filling pressures [138]. The latter finding was, however, only present in female HFpEF patients [138], supporting the concept that sex differences in fat tissue properties play a role in HFpEF [4], which is reviewed in detail elsewhere [139,140]. Nonetheless, direct observations between EAT or perivascular adipose tissue and microvascular function in HFpEF patients have not been published to date. Thus, future research is warranted for exploring the mechanism between MVD, adipose tissue, and inflammation in HFpEF.

### 4.2. MVD Pathways and Cardiomyocyte Stiffness

MVD has been proposed to play a role in cardiac fibrosis and cardiomyocyte stiffness in HFpEF [7]. A number of studies have displayed increased collagen deposition, cardiomyocyte hypertrophy with titin hypophosphorylation (a large spring-like molecule responsible for passive and active forces of cardiomyocytes), and vascular SMC changes (altered myosin subunits) in association with MVD in human HFpEF samples [8,12,108,141,142]. Franssen et al. reported elevated inflammation markers and a disturbed NO-cyclic guanosine monophosphate (cGMP)-protein kinase G (PKG)-signalling pathway in human cardiac biopsy samples from HFpEF patients as compared to patients with severe aortic stenosis or HFrEF, suggesting an impaired coronary microvascular function [8]. These findings have been replicated in several animal models [143,144,145]. Hence, it was postulated that MVD in HFpEF is due to inflammation and oxidative stress [7]. Supporting that finding, co-culture studies of cardiac microvascular endothelial cells and cardiomyocytes showed that endothelium-derived NO enhances cardiomyocyte contraction and relaxation, and that inflammation and oxidative stress can impair this endothelial function [146]. Serum from patients suffering from chronic kidney disease also inhibited this endothelial control of cardiomyocyte function, demonstrating clinical relevance [116]. These data suggest a potential causal role for the NO-cGMP-PKG pathway in HFpEF patients, but more human data are warranted. Alternatively, other underlying mechanisms, such as increased sympathetic nervous activity in HFpEF [147], together with high hypertension rates [148], and/or impaired glucose metabolism [149,150,151], may relate to MVD as a driver for HFpEF. These mechanisms also require more research. Given the phenotypic heterogeneity of MVD in HFpEF patients, including different endothelium-dependent and endothelium-independent alterations, other signalling pathways may play an important role in specific HFpEF patients. One of such pathways may be reduced X-box-binding protein 1 due to increased inducible NO synthase (iNOS) found in rodents as well as cardiac biopsies of HFpEF patients compared to controls without HF [152].

## 5. Microvascular Interventions in HFpEF

The above-described microvascular alterations associated with HFpEF have been instrumental in suggesting therapeutic targets based on underlying mechanisms. Several clinical trials have been performed targeting a variety of endothelial and cardiomyocyte-based proteins that are directly or indirectly aimed at tackling (the consequences of) MVD (Figure 2) [153,154,155,156,157]. Remarkably, the patients in these trials have not been selected for the presence of MVD or specific underlying microvascular mechanisms. Therefore, while the results of these trials can provide additional insights into the role of MVD in HFpEF, they should be interpreted carefully due to nonspecific patient selection for MVD presence. 

A large number of recent studies and ongoing trials have focused on restoring parts of the NO-sGC-cGMP-PKG pathway [158], particularly in the heart, to eventually improve the impaired PKG signalling that is reported to contribute to HFpEF [8] (Figure 2). We will briefly discuss the effects of sodium-glucose co-transporter-2 (SGLT2) inhibitors, soluble guanylate cyclase (sGC) stimulators, sacubitril-valsartan, phosphodiesterase (PDE) inhibitors, nitrates, and exercise training.

Interventions in HFpEF targeting MVD directly or indirectly probably provide most benefits when MVD is present, but likely also when interventions have a positive effect on several aspects of the multimorbid cardiovascular system of such patients. Considering the high prevalence of glucose intolerance and other metabolic comorbidities in HFpEF and their detrimental effect on microvascular function [149,159,160], this comorbidity burden is an important target in HFpEF treatment [146,161]. The EMPA-REG and subsequent EMPEROR-PRESERVED trials were breakthroughs in the treatment of HFpEF [162,163]. The antidiabetic drug Empagliflozin improved the composite endpoint of HF hospitalization or cardiovascular death in HFpEF patients with diabetes mellitus, but also in those without diabetes mellitus [163]. This drug has pleiotropic effects, but the improvement in the NO-sGC-cGMP-PKG pathway and reduction in inflammatory and oxidative stress contribute to its positive results, with enhancement of NO activity as one of the underlying mechanisms [146,164]. Remarkably, most other drugs targeting this pathway have shown, for the most part, neutral results in HFpEF. A summary of the conducted clinical trials is provided in Appendix A. 

Direct stimulators of sGC, such as vericiguat or praliciguat, have the capacity to activate the cGMP pathway independently from NO. A previous trial with vericiguat in HFpEF did lead to a decrease in self-reported physical limitations but not to changes in NT-proBNP or left atrial size [153]. However, the follow-up duration was relatively short, and not all patients were uptitrated to a higher or maximal dose, decreasing the potential effect of the intervention.

Sacubitril-valsartan, a combination of a neprilysin-inhibitor and an angiotensin-receptor blocker, enhances natriuretic peptide bio-availability, which influences microvascular function independent of NO, but failed to show a positive effect on HF hospitalization rates or death from cardiovascular causes in a general HFpEF population [165]. Nevertheless, an analysis of predefined subgroups suggested benefits in females [165], who make up a large proportion of the HFpEF population.

Phosphodiesterase (PDE)-5 inhibition, an established strategy to increase PKG activity, has shown conflicting results in HFpEF. A preclinical study showed protective effects of sildenafil on diastolic function by alleviating nitrosative stress in diabetic rats [166], but clinical trials have mostly shown neutral results in HFpEF patients [156,167,168]. 

Physical activity can induce skeletal muscle cells to become hypoxic, particularly when oxidative stress is present, limiting the muscle cells function [169]. This could be overcome by restoring NO signalling through nitrates and nitrites supplements, thus possibly improving exercise capacity [170,171]. As exercise intolerance is an important symptom of HFpEF patients, the effects of inorganic nitrates or nitrites in HFpEF seemed promising. This was supported by a few small trials using a single administration of inorganic nitrates or nitrites that showed improvement in exercise capacity [170] and cardiac haemodynamics (pulmonary capillary wedge pressure) [172] in HFpEF. Unfortunately, when administered for four weeks in a larger trial, the inorganic nitrites did not yield similar benefits [173]. Organic nitrates such as isosorbide mononitrate and dinitrate are known to cause side-effects like nitrate-induced systemic hypotension, which can be particularly limiting in HFpEF [174]. A trial that investigated the effect of isosorbide mononitrate even showed a decrease in daily activity compared to placebo, although they did not report major side effects [157]. 

In general, most trials did not include invasive cardiac measurements or peak exercise testing to evaluate the effects of the intervention. On the one hand, this could limit the ability of a trial to show an effect of the drug, especially in patients in earlier stages of HFpEF who present with fewer impairments regardless of intervention. On the other hand, studies using peak exercise or invasive methods may possibly have a selection bias by excluding frailer patients who are physically not able to participate. Moreover, aside from the nonspecific selection of HFpEF populations, most targeted trials were not powered to evaluate the effects of the intervention based on important MVD drivers.

Finally, regardless of MVD or HFpEF phenotype, physical activity in HFpEF can improve exercise capacity by improving factors such as chronotropic incompetence (insufficient heart rate increase despite higher cardiac output demands) and increasing oxygen use by active muscles, which improves the symptom burden [175]. Factors associated with MVD that improve with physical activity are increased endothelium-dependent vasoreactivity and an increase in peak arterial-venous oxygen difference [31,74,176,177,178]. It is hypothesized that the improvement in microvascular function seen after exercise training in HFpEF results from a wide range of improvements in organ function, including the lungs, kidneys, and the immune system [179]. This could explain why pharmacological interventions that have focused on one pathway have mostly yielded neutral results. Future studies on increased physical activity in HFpEF are warranted to define which HFpEF phenotype and disease stage can specifically benefit from this intervention.

## 6. Challenges

Currently, both MVD and HFpEF are still developing in terms of definitions and diagnosis. The number of studies has been increasing substantially, but studies to date have often provided incomplete knowledge and were difficult to compare with other studies, limiting firm conclusions. In addition, several important challenges regarding MVD and HFpEF include implications of different underlying mechanisms, the association between coronary and peripheral MVD, and confounding factors. The major limitations and knowledge gaps are summarized in Table 4.

MVD can be caused by different underlying mechanisms, including inflammation, oxidative stress, increased sympathetic activity, impaired glucose metabolism, and cellular and metabolic changes [59,82,181]. Knowledge of mechanisms underlying MVD could be crucial to guide future targeted HFpEF therapies, but human data from biological samples and prospective data are scarce. Animal models can help to generate hypotheses about these underlying mechanisms, even though most existing experimental models do not yet reflect the complex interconnections between HFpEF and MVD in humans, such as multifactorial impairments in elderly humans versus singular impairments in young/middle-aged mice that may lead to HFpEF [180].

Moreover, the association between coronary and peripheral MVD in HFpEF patients and its implications remains to be further investigated. Only one study assessed this association and used different stimuli for vasoreactivity in the coronary and finger vasculature [10].

In addition, it should be acknowledged that MVD is not specific to HFpEF, but it is associated with multiple diseases and cardiovascular risk factors. Data to date lack clear causative factors (determinants) for HFpEF or MVD in HFpEF, but associative factors (correlates) for MVD in HFpEF have been reported. Meanwhile, determinants of MVD outside the scope of heart failure have been discovered. These clinical factors are summarized in Table 3. Determinants of MVD include age [70,75], blood pressure [85], physical inactivity [89,91], obesity [93], hypercholesterolemia [71,81], and (pre)diabetes mellitus [17,72,82]. Sex and hormonal status, oestrogen and oestrogen receptor activity in particular, have also been found to be important factors influencing microvascular function and NO production [76,77,78,94,95,182,183,184]. In addition, given that HFpEF is predominantly present in post-menopausal women, sex differences and hormonal status may be important to improve insights into how these factors influence MVD and HFpEF. Along the same lines, microvascular results can be influenced acutely by dietary intake, such as caffeine and nutrients [86,87,88,185]. These aforementioned correlates and determinants are likely contributors to progress from a healthy individual to one with MVD and subsequent HFpEF. However, these correlates and determinants can influence both MVD and HFpEF, and in a bidirectional manner. For instance, physical inactivity leads to MVD, MVD leads to HFpEF, and chronic HF leads to microvascular remodelling and dysfunction (such as secondary pulmonary hypertension in HF [186]) and impaired physical capacity [187]. These interactions create a complex web of potential causal mechanisms (Figure 3). Moreover, these correlates potentially also contribute to HFpEF through different mechanisms than MVD.

## 7. Future Perspectives

Many different microvascular abnormalities have been found in selected HFpEF populations and were associated with worse outcome [9,24,25]. Both coronary and peripheral MVD were associated with incident and current HFpEF and adverse disease progression, suggesting a role of systemic MVD in HFpEF rather than just coronary MVD (Table 1 and Table 2) (Figure 4). Although diverse in measurements, the microvascular abnormalities found are relatively consistent in the available literature. Still, the number of studies is limited, and the specificity of microvascular abnormalities for HFpEF has not been addressed sufficiently to date due to inadequate considerations of other correlates for MVD and HFpEF (Table 4). The findings possibly reflect different HFpEF phenotypes and underlying mechanisms [150], but can also be a result of different techniques and vasoactive stimuli. Current findings generate the hypothesis that improving MVD could ameliorate HFpEF, yet testing this requires solutions to certain challenges and knowledge gaps mentioned in this review. Firstly, more clinical and large cohort studies are necessary to confirm the causal relation of MVD with HFpEF in a complex web of causal mechanisms. Still, causative and treatable factors can be found in this complexity, similar to the search for causality in low-density lipoprotein and atherosclerotic cardiovascular disease research [188]. It will require that reported associations are replicated by uniform measurements, temporal sequences of correlates and HFpEF are studied, potential confounders are taken into account, and interventions with a primary or secondary target on underlying mechanisms of systemic MVD in HFpEF are continued to be evaluated. Furthermore, studies on HFpEF and microvascular function should take the potential confounding comorbidities (listed in Table 3) into account in multivariable analyses, including interaction analyses, in order to properly evaluate their true contribution to HFpEF and to find subgroups of patients that could benefit from certain therapies.

Moreover, assessing MVD can also aid in identifying patients at higher risk for developing HFpEF, as current data suggest structural and functional signs of MVD occur already at a presymptomatic phase of HFpEF in different microvascular beds [37,109,120,189]. It is, however, not clear from which stage of MVD patients would develop HFpEF due to cumulative exposure and if certain thresholds of microvascular dysfunction exist in the development of HFpEF. In addition, the added value of MVD assessments in diagnosing or predicting HFpEF should be studied to find meaningful cut-off values in the MVD continuum. To achieve these goals, future studies should incorporate microvascular assessments to specifically assess relevant microvascular functions that are found systemically, such as vasoreactivity induced by acetylcholine and sodium nitroprusside. This requires more studies to compare coronary and peripheral microvascular functions in HFpEF to better understand systemic MVD in different vascular beds in relation to disease development and progression. In addition, since peripheral assessments allow easy access to microvascular function tests and are less challenging than coronary microcirculation assessments in terms of patient burden, costs, invasiveness, and required training, peripheral microvascular assessments can be performed in earlier phases of the disease and in larger populations [159,190]. For example, the skin is a suitable vascular bed for this purpose, because it allows to easily evaluate microvascular changes induced by multiple microvascular responses in the same location. These include microvascular rarefaction, diameter, capillary recruitment, and transcutaneous acetylcholine or sodium nitroprusside responses. Furthermore, incorporating targeted peripheral microvascular assessments in clinical trials would allow cause-effect analysis of therapies and outcomes in HFpEF with MVD as a secondary endpoint, which could greatly enrich knowledge about MVD and HFpEF and its therapeutic consequences. Finally, clinical studies assessing microvascular function or structure should also try to assess features of underlying mechanisms for MVD when biological samples are available. 

In order to answer some knowledge gaps, we have started to prospectively investigate microvascular changes of multiple vascular beds in HFpEF patients and controls (Netherlands Trial Register NL6428, NL7059, NL7655).

## 8. Conclusions

Systemic MVD is present in HFpEF, based on interpretation of abundant data from many correlational studies that show impairments in microvascular function, both endothelium-dependent and endothelium-independent, in different vascular beds. MVD should be seen as a continuum between function and dysfunction, which can influence HFpEF and comorbidity progression, and vice versa. Hitherto, due to a lack of clear causative evidence, it remains unknown how systemic MVD could drive HFpEF. 

Furthermore, HFpEF patients unequally show different elements of MVD, which might reflect different underlying mechanisms and therapeutic targets. Future research on MVD and HFpEF is, therefore, needed to uncover the true diagnostic and therapeutic value of microvascular assessments. This will require more uniformity and confounder considerations in study design, analyses, and reporting. However, the incorporation of peripheral microvascular assessments is feasible and should be considered in clinical HFpEF trials.

## Figures and Tables

**Figure 1 biomolecules-12-00278-f001:**
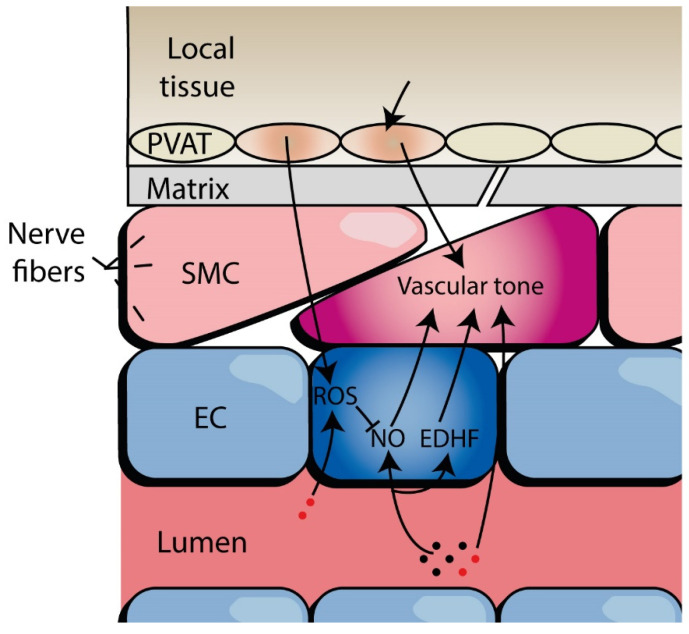
Example of intercellular signalling in an arteriole. A simplified example of the complex intercellular signalling of the microcirculation is displayed. This signalling is different per vessel type. It comprises a variety of systemic and local signals from surrounding tissue and the blood, acting in both the short- and long-term, including humoral, physical, neurogenic, cellular, and metabolic factors. Alterations in these signalling pathways and cellular abnormalities have been reported in HFpEF patients, including changes in matrix cell types and stiffness; adipose tissue cell phenotype and adipokine secretion; muscle cell hypertrophy and oxidative stress, and vasodilator response; endothelium-dependent vasodilation; microvascular rarefaction, and microvessel morphology. EC, endothelial cells; EDHF, endothelium-derived hyperpolarizing factors; NO, nitric oxide; PVAT, perivascular adipose tissue; ROS, reactive oxygen species; SMC, smooth muscle cells.

**Figure 2 biomolecules-12-00278-f002:**
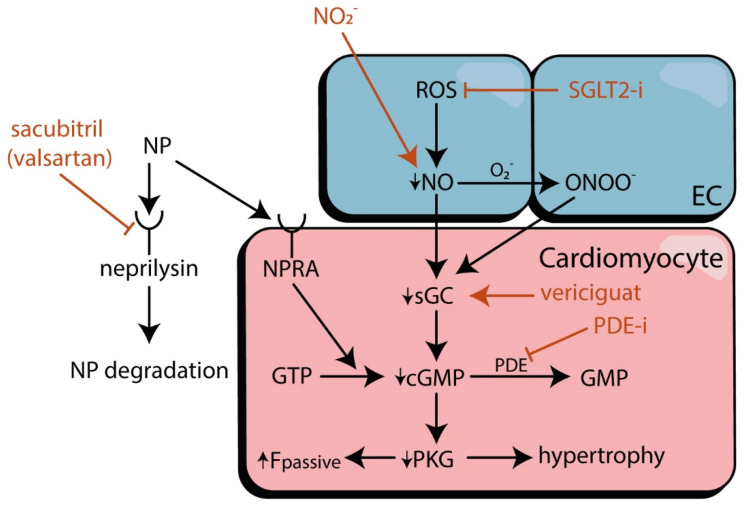
Interventions targeting MVD in HFpEF. The effects of different pharmacological interventions (blue) on the NO-sGC-cGMP-PKG pathway in MVD in HFpEF. ROS causes impaired NO bio-availability, subsequently disturbing the downstream signalling. The entire pathway offers different targets for therapy. cGMP, Cyclic guanosine monophosphate; EC, endothelial cell; Fpassive, passive force; NO, nitric oxide; NO_2_^−^, nitrite; NP, natriuretic peptides; NPRA, Natriuretic peptide Receptor Type A; O_2_^−^, superoxide; ONOO^−^, peroxynitrite; PDE-i, phosphodiesterase inhibitors; PKG, protein kinase G; ROS, reactive oxygen species; sGC, soluble guanylate cyclase; SGLT2-i, sodium-glucose co-transporter-2 inhibitor; SMC, smooth muscle cell.

**Figure 3 biomolecules-12-00278-f003:**
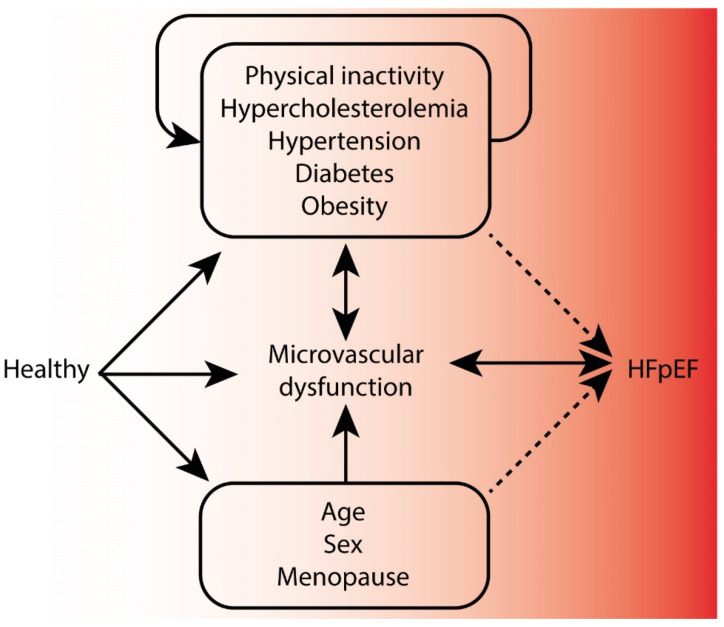
The potential causal web of clinical correlates for MVD and HFpEF. Clinical correlates affect both microvascular dysfunction (MVD) and heart failure with preserved ejection fraction (HFpEF), and in a bi-directional manner, creating a complex causal web. Yet, these clinical factors could also influence HFpEF development and progression through other mediating mechanisms (dotted lines) that are to be further elucidated. Moreover, the specificity of microvascular abnormalities for HFpEF requires further research. Causality of the clinical factors with systemic MVD and how this would drive HFpEF is still a knowledge gap.

**Figure 4 biomolecules-12-00278-f004:**
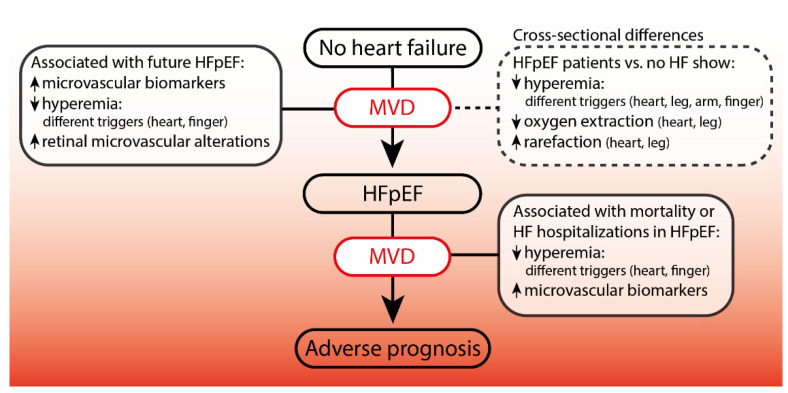
Findings of MVD and their associations within HFpEF. The presence of systemic microvascular dysfunction (MVD) in heart failure with preserved ejection fraction (HFpEF) is suggested by different clinical studies that show (1) cross-sectional microvascular differences between HFpEF and controls, (2) associations between baseline measures of microvascular function and incident HFpEF, and (3) associations between baseline microvascular measures in HFpEF patients and adverse disease progression defined by all-cause mortality or heart failure (HF) hospitalization. Due to current evidence limitations, causality between MVD and HFpEF requires further research.

**Table 1 biomolecules-12-00278-t001:** Studies on peripheral microvascular function in HFpEF.

Study Design	HFpEF Population	Control Population	Method (Measurement)	Stimulus	Microvascular Function Assessed	Outcome (SD/IQR)
Skin-finger						
Prospective [24]	*n* = 321	Controls without HF, matched for age, sex, HT, and DM (*n* = 173)	Peripheral arterial tonometry (endoPAT): (RHI)	Ischemia	Hyperaemia	Log RHI: 0.53 ± 0.20 vs. 0.64 ± 0.20, *p* < 0.001
Prospective [10]	*n* = 202	No controls	endoPAT (RHI)	Ischemia	Hyperaemia	Log RHI: no absolute values reported. Correlation with CFR of R 0.21, *p* = 0.004
Retrospective [25]	*n* = 159	No controls	endoPAT (RHI)	Ischemia	Hyperaemia	Log RHI: 0.50 ± 0.09. Event free 0.52 ± 0.09 vs. Events 0.46 ± 0.08, *p* < 0.001
Prospective (cross-sectional) [26]	*n* = 62	Controls matched for age, sex, HT, DM, dyslipidaemia and CAD (*n* = 64)	endoPAT (RHI)	Ischemia	Hyperaemia	RHI: 2.01 [1.64–2.42] vs. 1.70 [1.55–1.88], *p* < 0.001
Prospective [27]	*n* = 42	HFrEF (*n* = 46)	endoPAT (RHI)	Ischemia	Hyperaemia	RHI: 1.77 [1.67–2.16] vs. 1.53 [1.42–1.94], *p* = 0.014.
Prospective [28]	*n* = 26	Healthy controls, matched for age and sex (*n* = 26)	endoPAT (RHI)	Ischemia	Hyperaemia	RHI interpretation from boxplots: 1.9 [1.6–2.9] vs. 1.8 [2.0–3.3], *p* = 0.036. No effect of exercise
Prospective [29]	*n* = 21	HT controls without HF (*n* = 19) Healthy controls (*n* = 10)	endoPAT (RHI)	Ischemia	Hyperaemia	Log RHI: 0.85 ± 0.42 vs. 0.92 ± 0.38 vs. 1.33 ± 0.34, *p* = n.s. between HFpEF and HT controls
Skin-arm						
Prospective [30]	*n* = 45	HT controls, matched for age, sex and diabetic status (*n* = 45)	Laser Doppler flowmetry (LDF), power spectral density (PSD) of the LDF signal	None, ischemia	Vasomotion, hyperaemia	LDF PSD: lower in HFpEF, no absolute numbers reported, *p* < 0.05. Peak blood flow (PU): 135 [104–206] vs. 177 [139–216], *p* = 0.03
Prospective [11]	HFpEF with CAD *n* = 12	HFrEF with CAD (*n* = 12) CAD without HF (*n* = 12)	Laser Doppler imaging (LDI) coupled with transcutaneous iontophoresis of vasodilators	acetylcholine, sodium nitroprusside	Hyperaemia	Vasodilation due to Acth: No absolute values reported. *p* = 0.00099 (HF vs. controls). Vasodilation due to nitroprusside: *p* = 0.006 (HF vs. controls)
Muscle-leg						
Prospective [16]	*n* = 22	Healthy controls, age-matched (*n* = 43).	Histology (skeletal muscle biopsy of thigh)		Capillary density	Capillary-to-fibre ratio: 1.35 ± 0.32 vs. 2.53 ± 1.37, *p* = 0.006
Prospective [31]	*n* = 7	No controls.	Near-infrared spectroscopy: index for skeletal muscle haemoglobin oxygenation of thigh		Diffusion	Muscle deoxygenation overshoot was decreased after priming exercise, *p* = 0.041

Abbreviations: CAD, coronary artery disease; CFR; coronary flow reserve; DM, diabetes mellitus; HF, heart failure; HFpEF, heart failure with preserved ejection fraction; HFrEF, heart failure with reduced ejection fraction; HT, hypertension; MVD, microvascular disease; RHI, Reactive hyperaemia index.

**Table 2 biomolecules-12-00278-t002:** Studies on coronary microvascular function in HFpEF.

Study Design	Study Population	Method (Measurement)	Stimulus	Microvascular Function Assessed	Outcome (SD/IQR)	Outcome Adjusted for Confounders
Heart-autopsy						
Retrospective [12]	Deceased: HFpEF (*n* = 124); Controls (no HF) (*n* = 104)	Histology: microvessels/mm^2^ (microvascular density)		Rarefaction	Microvascular density: 961 (800–1370) vs. 1316 (1148–1467), *p* < 0.0001	Not performed, unmatched population
Invasive coronary function assessment						
Retrospective [14]	CAG after positive stress test: HFpEF > 65 (*n* = 32); HFpEF < 65 (*n* = 24); Controls (*n* = 31)	Invasive CFR and IMR	Adenosine	Hyperaemia	CFR: 1.94 ± 0.28 vs. 1.83 ± 0.32 vs. 3.24 ± 1.11, *p* ≤ 0.04 IMR: 39.2 ± 6.8 vs. 27.2 ± 6.4 vs. 18.3 ± 4.4, *p* ≤ 0.03	Age, sex, HT, DM, CKD, AF, BMI, LVMI. Unmatched controls
Retrospective [9]	HFpEF (*n* = 162)	Invasive CFR and coronary blood flow (CBF)	Adenosine, acetylcholine	Hyperaemia	No absolute values reported. Mortality is increased in coronary MVD (HR 2.8–3.5).	Age, sex, BMI, DM, HT, hyperlipidaemia, smoking, Hb, creatinine, uric acid
Retrospective [32]	HFpEF (*n* = 22); no HFpEF (*n* = 29)	Invasive CFR and CBF	Adenosine, acetylcholine	Hyperaemia	CFR: 2.5 ± 0.6 vs. 3.2 ± 0.7, *p* = 0.0003 Median CBF % increase: 1 (−35;34) vs. 64 (−4;133), *p* = 0.002	Age, sex
Prospective [33]	HFpEF with obstructive epicardial CAD (*n* = 38); HFpEF without epicardial CAD (*n* = 37)	CAG (CFR, coronary reactivity, IMR) and MRI	Adenosine, acetylcholine	Hyperaemia	CFR: 2.0(1.2–2.4) vs. 2.4(1.5–3.1), *p* = 0.06. IMR: 18(12–26) vs. 27(19–43), *p* = 0.02. 24% microvascular spasm due to Acth.	Clinical characteristics are compared between groups based on coronary results.
Prospective (cross-sectional) [13]	Clinical indication for CAG: HFpEF (*n* = 30); Controls (*n* = 14)	Invasive CFR and IMR	Adenosine	Hyperaemia	CFR: 2.55 ± 1.60 vs. 3.84 ± 1.89, *p* = 0.024 IMR: 26.7 ± 10.3 vs. 19.7 ± 9.7, *p* = 0.037	Exploratory analysis on age, BMI, GFR, BNP, echocardiographic data, hemodynamic data. Unmatched controls
Retrospective [34]	Patients with angina presented to the ER: HFpEF (*n* = 155); Controls (*n* = 135)	Total myocardial blush grade score (TMBGS)	None, nitroglycerin	Blood flow	TMBGS: 5.6 ± 1.22 vs. 6.1 ± 1.26, *p* = 0.02	Not performed, unmatched population
Non-invasive coronary assessment						
Prospective [35]	HFpEF (*n* = 19); Matched healthy controls (*n* = 19)	PET (C-acetate-11): myocardial blood flow (MBF) and myocardial oxygen consumption (MVO_2_)	Dobutamine	Blood flow, hyperaemia, diffusion	MBF increase: 78% vs. 151%, *p* = 0.0480 MVO2 increase: 59% vs. 86%, *p* = 0.0079 Absolute values during stress test not significantly different.	LVH, Hb. Healthy controls were matched for age and sex.
Retrospective [36]	Indication for cardiac PET: HFpEF (*n* = 78); HT without HF (*n* = 112); No HF no HT (*n* = 186)	PET (Rb-82): global myocardial flow reserve (MFR)	Dipyridamole	Hyperaemia	MFR: 2.16 ± 0.69 vs. 2.54 ± 0.80 vs. 2.89 ± 0.70, *p* ≤ 0.001	Age, sex, BMI, smoking, DM, HT, hyperlipidaemia, HT, AF, statin use. Controls matched for HT.
Retrospective [37]	Suspected CAD: Cohort without HF (*n* = 201)	PET (Rb-82): (CFR)	Regadenoson or dipyridamole	Hyperaemia	18% of the patients had a HFpEF event during follow-up. Independent HR with CFR <2.0 of 2.47 (1.09–5.62)	In entire cohort: AF, CKD, troponin, LVEF, CFR, E/e’ septal
Prospective [38]	HFpEF (*n* = 25); LVH (*n* = 13); Controls (*n* = 18)	MRI (CFR)	Adenosine	Hyperaemia	CFR: 2.21 ± 0.55 vs. 3.05 ± 0.74 vs. 3.83 ± 0.73, *p* ≤ 0.002	BNP, LVEF, E/e’, LA dimension
Retrospective [39]	HFpEF without events (*n* = 137), with events (*n* = 26)	MRI (CFR)	Adenosine	Hyperaemia	CFR: 2.67 ± 0.64 vs. 1.93 ± 0.38	Not performed
Prospective [40]	HFpEF (*n* = 6); Post MI (*n* = 6); Healthy controls (*n* = 20)	MRI: intravascular volume of basal septum (IVV)	Gadofosveset	Permeability	IVV: 0.155 ± 0.033 vs. 0.146 ± 0.038 vs. 0.135 ± 0.018, *p* = 0.413	Not performed, unmatched controls
Prospective [10]	HFpEF (*n* = 202)	Echocardiography (CFR)	Adenosine	Hyperaemia	CFR: 2.13 ± 0.51	Age, sex, BMI, AF, DM, CAD, smoking, LV mass, 6MWT, KCCQ, urinary albumin-creatinine ratio. No controls.
Prospective [41]	HFpEF (*n* = 77); Healthy controls (*n* = 30)	Echocardiography (CFR)	Adenosine	Hyperaemia	CFR: 1.7 ± 0.2 (with MVD) vs. 3.1 ± 0.4 (no MVD) vs. 3.4 ± 0.3 (control)	Age, LAVI, LVMI, LVEF, E/e’, 6MWT distance

Abbreviations: AF, atrial fibrillation; BMI, body mass index; CAD, coronary artery disease; CAG, coronary angiography; CFR; coronary flow reserve; CKD, chronic kidney disease; CMD, coronary microvascular dysfunction; DM, diabetes mellitus; ER, emergency room; GFR, glomerular filtration rate; HF, heart failure; HFpEF, heart failure with preserved ejection fraction; HR, hazard ratio; HT, hypertension; IMR, index of microcirculatory resistance; LAVI; left atrial volume index; LV, left ventricle/ventricular; LVEDI, left ventricular end-diastolic volume indexLVEF, left ventricular ejection fraction; LVMI, left ventricular mass index; MFR, myocardial flow reserve; MVD, microvascular disease; PET, positron emission tomography; SR, sinus rhythm.

**Table 3 biomolecules-12-00278-t003:** Clinical factors associated with microvascular dysfunction.

Clinical Factor	Measurement Method	Microvascular Bed Assessed	Effect on Microvascular Function
Age [36,70,73,74,75]		Skin, eye, skeletal muscle, heart	Function decreases by increasing age
Hormonal status [76,77,78,79]	Oestrogen levels, together with oestrogen receptor activity, are most accurate. Menopausal status and oral contraceptive therapy use are alternative surrogate markers.	Skin, skeletal muscle, heart	Function decreases with lower oestrogen activity
Hypercholesterolemia [71,80,81]	Serum cholesterol panel	Skin, eye, heart	Function decreases with higher serum low-density lipoprotein cholesterol levels
Hyperglycaemia [82,83]	Glucose tolerance test, fasting glucose, HbA1c	Skin, eye, heart	Function decreases with higher plasma glucose levels
Hypertension [36,38,70,84,85]	24-h systolic blood pressure shows the highest correlation	Skin, eye, skeletal muscle, heart	Function decreases with higher systolic blood pressure and by duration of hypertension
Dietary intake [86]	Caffeine	Skin	Function is temporarily increased
Dietary intake [87,88]	High-fat diet	Skin, heart	Function is temporarily decreased
Physical inactivity [31,89,90,91]	24-h accelerometer, physical activity questionnaire	Skin, eye, skeletal muscle	Function decreases with more physical inactivity.
Obesity [8,70,92,93]	Waist circumference is more correlated than BMI or BSA.	Skin, eye, skeletal muscle, heart	Function decreases with increasing level of obesity
Sex [94,95]		Skin, eye, skeletal muscle, heart	Effect on function depends on other confounders.
Smoking [75,96]	Self-reported use	Skin, eye, heart	Function decreases with smoking and more pack years.

Abbreviations: BMI, body mass index; BSA, body surface area; HbA1c, glycated haemoglobin.

**Table 4 biomolecules-12-00278-t004:** Major limitations and knowledge gaps for MVD in HFpEF.

**Limitation**	HFpEF is a heterogeneous syndrome and has a variety of diagnostic criteria. This leads to different study populations with different disease stages and clinical phenotypes [5].
	Studies mainly focused on a general pathophysiology for an entire HFpEF population rather than selected phenotypes.
	The microvascular function is a continuum, and there is no clear cut-off value to define microvascular dysfunction in human biology. Nevertheless, studies are often focused on finding cut-off values that can have direct clinical implications. Both quantitative and categorical approaches are needed to improve current knowledge of MVD in HFpEF and could help in evaluating the likelihood of biological causality.
	No gold standard exists to assess MVD. Studies often overgeneralize specific microvascular alterations, which limits study designs and comparability within and between studies.
	MVD is not exclusive to HFpEF. Important drivers and clinical correlates of both (Table 3) need to be accounted for when interpreting study results. This requires larger study populations, which are not available in most of the current literature.
	Intervention studies with a direct or indirect effect on MVD have not selected patients based on those MVD aspects that are targeted with the intervention, limiting causal inference.
**Knowledge Gap**	Knowledge on causality and underlying mechanisms of MVD and HFpEF is mostly derived from animal models, but their agreement with corresponding human phenotypes needs more research [180].
	Comparable longitudinal data of microvascular function are lacking in healthy individuals, HFpEF development, and adverse disease progression due to heterogeneity in microvascular assessments and HFpEF definitions.
	The role of MVD in specific HFpEF phenotypes remains to be elucidated.
	The similarity and underlying mechanisms of peripheral and coronary MVD in HFpEF deserve more investigation, as similar impairments with the same underlying mechanism could guide future targeted therapies.

## Data Availability

Not applicable.

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
