# Peer review of "The Role of Systemic Microvascular Dysfunction in Heart Failure with Preserved Ejection Fraction"

_biomolecules, 2022, doi:10.3390/biom12020278_

Round 1

Reviewer 1 Report

The review is very complete and I appreciated the breadth of topics covered in this review article.  It is enjoyable to read. The authors review the methods used in clinical studies to assess whether MVD is present in HFpEF, biomarkers that may identify the presence of MVD, the role of adipose tissue and possible interventions to prevent MVD and rarefaction.  The review also astutely highlights many important potential components of HFpEF development such as sex differences in adipose tissues deposition.  The review shows the complexity and nuance of HFpEF and differences in populations affected by the disease.

I have only a few relatively minor corrections:

1) The introduction starts with the instructions for a journal format.  Was this an omission?  Or is it standard for this journal that during the peer review, the instructions still appear?

2) The authors state that Table 1 lists techniques in studies but then does not give the references for those studies. It seems this table is a bit of an introduction to table 2/3.  But then not all elements of table 1 are in table 2/3.  Some aspects only show up in the text several pages later.  It would be best to list reference, even if they are then expanded on in Table 2/3. 

3) It is also not clear until the reader gets to Table 2/3, what the point of Table 1 is.  Meaning, is this a list of clinically techniques or research techniques?  I would therefore make it clearer that these are techniques used in clinical research when referencing the table.

4) The retina is part of the central nervous system. It therefore seems inappropriate to list this under “peripheral”.  Is the brain or spinal cord a peripheral organ? Adding a section on “CNS” (which just lists retina) would avoid this problem.  Or choose a word other than “peripheral”.  Perhaps cardiac and non-cardiac?

5) Histology is referred to as “Immunohistochemistry” for the thigh muscle but as “Microscopy” for the cardiac tissue.  This should be made consistent.

6) Section 3 is “Evidence of microvascular dysfunction in HFpEF”. Most sections then discuss evidence that there is microvascular dysfunction present, but sections and 3.4 and 3.5 are more about the possible mechanism by which MVD would contribute to HFpEF. Section 3 could be renamed something more general such as “Microvascular vascular dysfunction in HFpEF”.

Minor

  • “MVD is not synonym for endothelial dysfunction ..” should be “is not a synonym…”
  • Hyperemia is defined in section 3.1.1 but first used in section 3.1. It should be defined on first use.

Reviewer 2 Report

The review by Weerts et al, “The role of systemic microvascular dysfunction in heart failure with preserved ejection fraction” is a very huge work written in a good English languish. However, too many papers are reported without a synthetic criticism. In addition, a clear method to categorize the current literature is lacking. Tables add confusion, in that too much information on each paper are reported, resulting in a boring reading. Pictures are otherwise of good quality; however, they are not well explained within the text, but almost exclusively in the relative legends.

Last but not least, the target readers of Biomolecules have various specializations and interests , so that the use of jargon is not indicated as for journals in the specific field of Cardiology.

Reviewer 3 Report

Weerts et al. reviewed the role of systemic microvascular dysfunction (MVD) in heart failure with preserved ejection fraction (HFpEF). Based on the interpretation of data from many correlational studies, the authors conclude that systemic MVD is present in HFpEF. The authors also point out that HFpEF patients unequally show different elements of MVD, which might reflect different underlying mechanisms and therapeutic target. Therefore, future research on MVD and HFpEF is needed to uncover the ture diagnostic and therapeutic value of microvascular assessments.

Overall, the review is well written. The figures are well drawn. There are only some minor concerns.

  1. The first paragraph of Introduction seems to be the “instruction for introduction”and should be deleted.
  2. The readability of Tables 2 and 3 is low. The Tables are too big and long. Some columns, such as “study’s first author”in Table 2, could be deleted.
  3. The relationship between peripheral, coronary, and systemic MVD should be discussed. Which type of MVD is more related to HFpEF?

Round 2

Reviewer 2 Report

The paper is still too long. Please reduces number of Tables. 

Author Response

We thank the reviewer for the feedback on our manuscript.

Point 1. “The paper is still too long. Please reduces number of Tables.”

Response 1. We acknowledge the length of the manuscript. We have moved the original Table 1 to the supplemental materials in the prior revision. Table 4 still adds information regarding clinical trials in addition to the text, particularly to readers with a clinical background, and has now been moved to supplemental materials as well. We believe the remaining four tables contain essential data and relevant context for the readers and have left these tables unchanged.